# Design, Synthesis, and Potent Anticancer Activity of Novel Indole-Based Bcl-2 Inhibitors

**DOI:** 10.3390/ijms241914656

**Published:** 2023-09-28

**Authors:** Ahmed M. Almehdi, Sameh S. M. Soliman, Abdel-Nasser A. El-Shorbagi, Andrew D. Westwell, Rania Hamdy

**Affiliations:** 1College of Sciences, University of Sharjah, Sharjah P.O. Box 27272, United Arab Emirates; ahmedm@sharjah.ac.ae; 2Research Institute for Science and Engineering (RISE), University of Sharjah, Sharjah P.O. Box 27272, United Arab Emirates; 3Research Institute for Medical and Health Sciences, University of Sharjah, Sharjah P.O. Box 27272, United Arab Emirates; ssoliman@sharjah.ac.ae; 4College of Pharmacy, University of Sharjah, Sharjah P.O. Box 27272, United Arab Emirates; aelshorbagi@sharjah.ac.ae; 5School of Pharmacy and Pharmaceutical Sciences, Cardiff University, Redwood Building, Cardiff CF10 3NB, UK; 6Faculty of Pharmacy, Zagazig University, Zagazig 44519, Egypt

**Keywords:** Bcl-2, anti-apoptotic protein, indole, cancer drug discovery

## Abstract

The Bcl-2 family plays a crucial role in regulating cell apoptosis, making it an attractive target for cancer therapy. In this study, a series of indole-based compounds, **U1**–**6**, were designed, synthesized, and evaluated for their anticancer activity against Bcl-2-expressing cancer cell lines. The binding affinity, safety profile, cell cycle arrest, and apoptosis effects of the compounds were tested. The designed compounds exhibited potent inhibitory activity at sub-micromolar IC_50_ concentrations against MCF-7, MDA-MB-231, and A549 cell lines. Notably, **U2** and **U3** demonstrated the highest activity, particularly against MCF-7 cells. Respectively, both **U2** and **U3** showed potential BCL-2 inhibition activity with IC_50_ values of 1.2 ± 0.02 and 11.10 ± 0.07 µM using an ELISA binding assay compared with 0.62 ± 0.01 µM for gossypol, employed as a positive control. Molecular docking analysis suggested stable interactions of compound **U2** at the Bcl-2 binding site through hydrogen bonding, pi-pi stacking, and hydrophobic interactions. Furthermore, **U2** demonstrated significant induction of apoptosis and cell cycle arrest at the G1/S phase. Importantly, **U2** displayed a favourable safety profile on HDF human dermal normal fibroblast cells at 10-fold greater IC_50_ values compared with MDA-MB-231 cells. These findings underscore the therapeutic potential of compound **U2** as a Bcl-2 inhibitor and provide insights into its molecular mechanisms of action.

## 1. Introduction

The B-cell lymphoma 2 (Bcl-2) protein is a key regulator of apoptosis, a programmed cell death process required for tissue homeostasis and the elimination of damaged cells [1]. Bcl-2 is a member of the Bcl-2 protein family, which is categorized into pro-apoptotic (such as Bax and Bak) and anti-apoptotic members (such as Bcl-2, Bcl-XL, and Mcl-1) [2]. Bcl-2 is an anti-apoptotic protein that suppresses cell death by inhibiting the activation of pro-apoptotic proteins, hence favouring cell survival [3]. Bcl-2 expression or function dysregulation contributes to cancer development and progression [4,5]. Several studies have linked the overexpression levels of the anti-apoptotic Bcl-2 protein with resistance to chemotherapy, radiation treatment, and targeted therapies in many cancer types [6,7]. This resistance permits cancer cell survival, resulting in therapeutic resistance and disease progression [8]. Furthermore, multiple Bcl-2 inhibitors are being tested in clinical studies against various cancer types [9,10], and activity has been widely reported in pre-clinical models such as human breast cancer cells MCF-7 [11,12] and MDA-MB-231 [13] and human lung cancer cell A549 [14,15]. These trials are assessing their safety, efficacy, and potential synergistic benefits when combined with existing medicines, offering the promise of improved cancer treatment outcomes [16].

The BH3 domain (Bcl-2 homology 3) is a conserved region present in pro-apoptotic Bcl-2 family proteins [17]. It facilitates the interactions of pro-apoptotic and anti-apoptotic proteins and is required for apoptosis regulation [18]. Small molecule inhibitors that mimic the BH3 domain and specifically bind to anti-apoptotic Bcl-2 proteins can disrupt this protein–protein interaction, resulting in apoptosis activation in cancer cells [19]. These inhibitors aim to sensitise cancer cells to apoptotic signals and enhance the effectiveness of cancer treatments [20]. Several BH3 mimetic molecules have been developed and tested as potential anticancer agents (Figure 1). Venetoclax (ABT-199) is an FDA-approved medication that targets Bcl-2 [21]; it is the first-in-class Bcl-2 inhibitor that treats lymphoid cancers [22] and has shown remarkable clinical activity against certain types of haematological malignancies such as chronic lymphocytic leukaemia [23]. Venetoclax combined with azacitidine is used for the treatment of acute myeloid leukaemia [24]. ABT-737 was also one of the first developed Bcl-2 inhibitors that demonstrated efficient activity against lymphoma [25]. However, its major limitation is low oral bioavailability; therefore, navitoclax (ABT-263) was developed [26]. Navitoclax (ABT-263) is a small-molecule Bcl-2 inhibitor with a broader inhibition profile of Bcl-2 family proteins [27]. It induces apoptosis in small-cell lung cancer and other solid tumour patients [28] and showed safety and efficacy when combined with erlotinib in the treatment of solid tumours [29]. Current Bcl-2 inhibitors, such as venetoclax, have limited applications because of the overexpression of anti-apoptotic proteins, particularly MCL-1, which leads to resistance and decreases in patient survival rates [30]. Obatoclax mesylate (GX15-070) is a small-molecule pan-Bcl-2 family inhibitor that is employed in the treatment of advanced chronic lymphocytic leukaemia [31]. Indole scaffolds play a crucial part in cancer cell targeting, particularly because the nitrogen atom in indole can form hydrogen bonds with biological targets, enhancing their potential activity [32,33,34]. Several research groups have demonstrated the incorporation of the indole scaffold and structural optimization to target cancer cells [34,35,36,37]. Obatoclax is an indole-based Bcl-2 inhibitor [38] that inspired the design of several indole-based compounds that bind directly to the Bcl-2 protein and suppress its anti-apoptotic action [39,40]. It also influences the mitochondrial apoptotic pathway by changing the permeability of the mitochondrial membrane, releasing pro-apoptotic proteins, and inducing apoptosis [41,42]. Other indole-based compounds have caused the production of reactive oxygen species [43] and caused cancer cells to enter cell cycle arrest [44], halting uncontrolled proliferation and triggering apoptosis [45].

As a result, this study attempted to identify novel inhibitors of the Bcl-2 protein that would provide enhanced therapeutic benefits by integrating distinct pharmacophoric moieties to improve potency, selectivity, and binding affinity towards target proteins. The development of new Bcl-2 inhibitors represents a promising therapeutic approach in oncology, with the potential to overcome drug resistance and improve the efficacy of cancer treatments. Continued research and clinical trials can reveal the full potential of Bcl-2 inhibition in cancer treatment.

## 2. Results

### 2.1. Rational Design

The rational design of Bcl-2 inhibitors has been explored using various small-molecule inhibitors with core heterocyclic scaffolds, such as isoquinoline [2,46] and indole [39,47], along with different substitutions, to enhance their activity and physiochemical properties. We previously designed a series of indole-based compounds with triazole [48], oxadiazole [3,49], and quinoline-fused triazolothiadiazole [2] scaffolds. The inclusion of a triazole ring was beneficial in targeting anti-apoptotic proteins Bcl-2 and Mcl-1 [45]. Morpholino substitution has demonstrated efficacy in targeting Bcl-2 proteins as in the case of ABT-263 [50,51]. Additionally, acetamide derivatives have been investigated for their ability to inhibit anti-apoptotic Bcl-2 proteins [52]. Various substitutions of electron-donating or -withdrawing groups have been introduced to the aforementioned compounds to alter the electronic properties and physicochemical characteristics of the molecule, potentially impacting its interactions with the target protein or modulating its biological activity. The rational construction of a series of molecules involves incorporating various functional groups and substitution patterns, including an indole ring, a 1,2,4-triazole ring, a thioether linkage, an acetamide linker, and a morpholino moiety. The arrangement and connectivity of these structural elements can contribute to the compound’s functional activity. It is worth noting that the discovery of small-molecule inhibitors targeting the Bim binding site of anti-apoptotic Bcl-2 is an active area of research (Figure 2). The rational design of these compounds aims to optimize their potency, selectivity, and other pharmacological properties, with the ultimate goal of developing effective therapeutic agents for the treatment of specific diseases.

### 2.2. Synthesis of Title Compounds ***U1**–**6***

A multistep procedure was used to synthesize compounds **U1**–**6**. First, 2-chloro-*N*-(4-nitrophenyl)acetamide **2** was synthesised by refluxing 2-chloroacetyl chloride in a dichloromethane solution of *p*-nitroaniline and potassium carbonate. Then, 2-morpholin-4-yl-*N*-(4-nitrophenyl)acetamide **3** was prepared by heating compound **2** in anhydrous THF with morpholine. Following that, *N*-(4-amino-phenyl)-2-morpholin-4-yl-acetamide **4** was synthesized by refluxing compound **3** in ethanol and HCl with reduced iron. Further, the reaction of compound **4** with 2 chloroacetyl chloride produced *N*-[4-(2-chloro-acetylamino)-phenyl]-2-morpholin-4-yl-acetamide **5**. The substituted-phenyl-5-(1H-indol-3-yl)-4H-1,2,4-triazole-3-thiols (**7a**–**f**) were synthesized by treating previously reported indolyl-3-carbonyl-*N*-substituted phenyl thiosemicarbazides (**6a**–**f**, 10 mmol) in refluxing sodium hydroxide solution [49]. Finally, compound **5** was mixed into ethanol and KOH with substituted-phenyl-5-(1H-indol-3-yl)-4H-1,2,4-triazole-3-thiol (**7a**–**f**) to effect the *S*-alkylation of the triazole thiol and yield the desired products, **U1**–**6**, as illustrated in (Figure 1). NMR spectra are available at Appendix A.

### 2.3. Compounds ***U2*** and ***U3*** Showed Potent Inhibitory Activity towards Bcl-2-Expressing Cancer Cells

The designed compounds demonstrated inhibitory activity against Bcl-2-expressing human cell lines, including breast cancer lines MCF-7 and MDA-MB-231 and A549 lung cancer cells at sub-micromolar concentrations, indicating their potency (Table 1, Figure 3). The most potent activity was observed in **U2**, followed by **U3**, **U1**, and **U4**. Overall, the antiproliferative activity of all compounds was highest against the MCF-7 cancer cell line. Compounds **U2** and **U3** showed IC_50_ values of 0.83 ± 0.11 and 1.17 ± 0.10 µM against MCF-7; 0.73 ± 0.07 and 2.98 ± 0.19 µM against A549; and 5.22 ± 0.55, 4.07 ± 0.35 µM against the metastatic and treatment-refractory triple-negative breast cancer cell line MDA-MB-231, respectively. The physicochemical characteristics of various substitution patterns and the potent activity of **U2** and **U3** led to their selection for further investigation.

### 2.4. ELISA Indicated the Superior Activity of Compound **U2** against Bcl-2 Protein

The ability of compounds **U2** and **U3** to bind to the Bcl-2 binding pocket as BH3 mimetics was tested with the ELISA binding assay, as previously described by our group [48,49]. The different attributes of the substitution, such as size, shape, and electrostatic interaction, influence their effectiveness as Bcl-2-competitive inhibitors. Compound **U2** exhibited an IC_50_ of 1.2 ± 0.02 µM, which is two-fold less potent than gossypol (IC_50_ 0.62 ± 0.01 µM), while **U3** exhibited lower binding affinity with an IC_50_ of 11.10 ± 0.07 µM (Table 2, Figure 4). These results indicated the potential activity of compound **U2** as a Bcl-2 inhibitor, warranting further investigation.

### 2.5. Compounds ***U1**–**6*** Showed an Excellent Safety Profile in Human Normal Cells

All tested compounds displayed minimal inhibitory activity against HDF human dermal normal fibroblast cells at a concentration of 50 µM (over 24 hrs), indicative of a high safety profile. In particular, compounds **U1**, **U3**, **U4**, and **U5** showed no significant change compared with the negative control. **U6** and **U2** exhibited limited toxicity on fibroblasts with *p*-values of <0.0001 and 0.01, respectively (Figure 5).

### 2.6. Compound ***U2*** Induced Apoptosis and Cell Cycle Arrest at G1/S Phase

Compound **U2** effectively inhibited cell growth through apoptosis and significantly increased early apoptosis, displaying a 43-fold increase compared with control untreated cells (Figure 6A–C), while inducing a remarkable 111-fold increase in late apoptosis (Figure 6A–C). Additionally, the compound caused cell cycle arrest at the G1/S phase, as indicated in Figure 7D–F. These findings highlight the ability of compound **U2** to induce programmed cell death and disrupt cell cycle progression.

### 2.7. Molecular Docking Revealed Efficacy and Selectivity of ***U2*** Compound against Bcl-2

Molecular docking analysis revealed that all **U1**–**6** compounds formed stable interactions at the Bcl-2 binding site (PDB: 4AQ3). The carbonyl group of **U2** showed H-bond interactions with Arg-60 with a distance of 3.16 A°, the phenyl ring showed pi-pi stacking interactions with Phe-63 alongside hydrophobic interactions with Ala-108, Arg-105, Glu-95, Tyr-67, and Phe-63, which further contributed to the binding stability (Table 3, Figure 7A). The carbonyl group of compound **U3** showed 2H-bond interactions with Arg-105 with distances of 2.94 and 2.97 A°, in addition to aromatic H-bond interactions between the phenyl ring and Val-92 with a distance of 2.70 A° and hydrophobic interactions with Arg-105, Glu-95, Asp-70, Leu-96, Tyr-67, and Phe-63 (Table 3, Figure 7B). On the other hand, the carbonyl group of compound **U1** showed 2H-bond interactions with Arg-105 with distances of 2.89 and 3 A° and aromatic H-bond interactions with Val-92 with a distance of 2.66 A°; the indole ring showed two aromatic H-bond interactions with Asp-99 and Glu-95 with distances of 2.75 and 2.46 A°, respectively, alongside hydrophobic interactions with Asp-70, Phe-63, Try-67, and Arg-105 within the binding site (Table 3, Figure 7C). The collective interactions resulted in an overall glide score of −5.8 kcal/mole, suggesting a strong binding affinity. Figure 7D–F show the 2D interaction maps of compounds **U2**, **U3**, and **U1**, respectively, within the Bcl-2 binding site.

### 2.8. Compound ***U2*** Showed Improved Properties Compared with Small-Molecule Inhibitor ABT263

The bioavailability radars (Swiss-ADME) of the newly designed and highly potent compound **U2** were compared with those of the ABT-263 compound (Figure 8). The pink areas in the figure represent the optimal range of six properties, namely, unsaturation, insolubility, lipophilicity, flexibility, polarity, and size. It was observed that the **U2** compound fell within the desired range and exhibited acceptable parameters when compared with ABT-236. This suggests that **U2** exhibited favourable bioavailability characteristics, making it a promising candidate for further development.

## 3. Discussion

Venetoclax, an FDA-approved Bcl-2 inhibitor, is quickly evolving as the standard of care for acute myeloid leukaemia and chronic lymphocytic leukaemia. Because of the concomitant upregulation of anti-apoptotic proteins, particularly MCL-1, following prolonged exposure to venetoclax [53], which provides an escape route leading to venetoclax resistance, its use is limited, and hence, it has become part of a combination therapy [54]. To overcome this limitation, researchers have studied the dual inhibition of MCL-1 and Bcl-2 [55,56]. This includes the use of an indole-based lead compound that exhibits potent MCL-1 inhibition and moderate Bcl-2 inhibition [57].

Indole-based compounds showed potent anticancer activity through the dual inhibition activity of both Bcl-2/Mcl-1 proteins [58]. Indole-based derivatives designed by incorporating amide, biphenyl, and sulphonamide pharmacophoric moieties aimed to enhance inhibitory activity against Bcl-2 and Mcl-1 [59], potentially leading to more effective dual inhibitors [58]. Furthermore, indole-based coumarin compounds have been reported to be potent anticancer agents by targeting Bcl-2, and halogen substitution has shown the highest activity [60]. We previously designed and synthesized a series of indole-based compounds by incorporating various heterocyclic rings, such as oxadiazole [61], triazole [62], thiazole [2,63], quinoline [2], and a fused triazolo-thiazole ring [64]. Our previous studies highlight the chloro-substitution of indole-based oxadiazole amine as a promising inhibitor of Bcl-2 with a similar binding affinity to gossypol; triazolo-thiazole and the dimethoxy derivatives showed two-fold less potent binding affinity to Bcl-2 compared with gossypol, employed as a positive control [64].

The structure modifications aimed to optimize the binding interactions between the inhibitor and the protein. The extension to occupy more of the binding pocket of Bcl-2 played a vital role in enhancing the inhibitory activity against both proteins. The design of new molecular features that enhance potency and specificity was achieved by studying the pharmacophoric features of potent inhibitors and structural characteristics to make favourable interactions. A molecular docking study indicated the binding affinity of the designed compounds with Bcl-2 active site key amino acid residues due to the amide carbonyl moiety incorporated with H-bond interactions. A structure–activity relationship study indicated that the substitution of electron-donating groups at the para-position, such as *p*-methoxy **U2** and *p*-methyl substituted **U3**, showed the highest activity, followed by electron-withdrawing functional groups such as *p*-fluoro-substituted **U1**. These modifications can impact the compound’s pharmacokinetic profile and potentially improve its efficacy and safety. A future priority will be to test the activity of the **U2** compound on anti-apoptotic members such as BCL-XL and MCL-1.

The **U2** compound showed cytotoxic effects mainly by inducing apoptosis and G1/S cell cycle arrest. This inference is consistent with earlier research highlighting the involvement of Bcl-2 inhibitors in cell cycle arrest, specifically in the G0/G1 and S phases. For example, TW-37, a small-molecule Bcl-2 inhibitor, induces S-phase arrest in tumours [65]. Obatoclax inhibited the G0/G1 cell cycle in human oesophageal cancer cells [66]. Other studies have found that Bcl-2 has the most pronounced effect on the cell cycle by delaying passage from the G0/G1 phase to the S phase [67] and that direct inhibition of Bcl-2 caused by obatoclax improves the G1/G0 to S phase transition rather than causing G1/G0 phase arrest [68]. Gossypol also limited cell cycle progression by triggering S phase arrest [69]. These findings contribute to continuing attempts to find more effective Bcl-2 inhibitors, perhaps leading to new therapeutic alternatives for cancer treatment. A future study will be carried out to compare the activity of our designed compound compared with further positive control compounds in the cell cycle and apoptosis assays. More study and optimization of these compounds are required to increase their potency and selectivity.

## 4. Experimental Section

### 4.1. Chemistry

The synthesised compounds were monitored with thin-layer chromatography (TLC) using pre-coated silica gel plates (Kieselgel 60F254, BDH, Taufkirchen, Germany) and visualised using UV light at 254 nm. A Gallenkamp melting point apparatus (London, UK) was used to measure the melting points (mps). ^1^H NMR spectra were acquired at 500 MHz with a Bruker spectrometer. Chemical shifts were indicated in parts per million (ppm) relative to TMS; coupling constant (J) values were expressed in hertz (Hz); and the signals were labelled as s (singlet), d (doublet), t (triplet), and m (multiplet). Positive mode electrospray ionization (ESI) mass spectroscopy (Bruker Daltonics mass spectrometer, Bremen, Germany) was used to confirm molecular mass and formula.

#### 4.1.1. Synthesis of 2-Chloro-*N*-(4-nitrophenyl)acetamide (**2**)

Chloro-acetyl chloride (1 mL, 10 mmol) was added dropwise to a suspension of *p*-nitro aniline (1.4 g, 10 mmol) and potassium carbonate (20 mmol) in DCM (20 mL). The reaction mixture was refluxed for 4 h. The aqueous layer was extracted with dichloromethane, dried over anhydrous sodium sulphate, and evaporated in vacuo until dry. The crude product was recrystallized from ethanol. Yield: 70%, mp: 158–159 °C. ^1^H-NMR (DMSO-d_6_) δ 4.32 (s, 2H, CH_2_), 7.83 (d, 2H, J = 9.15, ArH), 8.25 (d, 2H, J = 9.15, ArH), 10.38 (s, 1H, NH).

#### 4.1.2. Synthesis of 2-Morpholin-4-yl-*N*-(4-nitrophenyl)acetamide (**3**)

2-Chloro-*N*-(4-nitrophenyl) acetamide (**2**, 2 g, 10 mmol) in anhydrous THF (10 mL) was added dropwise to K_2_CO_3_ solution (2.8 g, 20 mmol), followed by the addition of morpholine (**1**, 1 mL, 10 mmol). The reaction mixture was heated to 80 °C for 16 h. Water (30 mL) was added, and the resulting precipitate was collected via filtration, rinsed with water, and allowed to dry to obtain the crude product without further purification.

#### 4.1.3. Synthesis of *N*-(4-Amino-phenyl)-2-morpholin-4-yl-acetamide (**4**)

2-Morpholin-4-yl-*N*-(4-nitrophenyl)acetamide (**3**, 2.7 mL, 10 mmol) in 75% ethanol (30 mL) was added to a solution of reduced iron 5 gm in water (10 mL) and concentrated hydrochloric acid (0.5 mL) solution. The reaction mixture was heated to reflux for 1 h. Iron was removed through filtration, and the filtrate was rinsed three times with ethanol. The filtrate was then run through a silica pad to eliminate any leftover iron residues. The filtrate was evaporated in vacuo until dry and purified using preparative TLC. Yield: 92%, mp: 97–99 °C. ^1^H-NMR (DMSO-d_6_) δ 2.49 (s, 4H, 2 × CH_2_), 3.01 (s, 2H, CH_2_), 3.66 (m, 6H, 2 × CH_2_, NH_2_), 6.54 (d, 2H, J = 8.65, ArH), 7.22 (d, 2H, J = 8.65, ArH), 8.78 (s, 1H, NH).

#### 4.1.4. *N*-[4-(2-chloroacetylamino)phenyl]-2-morpholin-4-yl-acetamide (**5**)

Chloroacetyl chloride (1 mL, 10 mmol) in anhydrous THF (10 mL) was added dropwise to K_2_CO_3_ solution (2.8 g, 20 mmol), followed by *N*-(4-amino-phenyl)-2-morpholin-4-yl-acetamide (**4**, 2.4 mL, 10 mmol). The reaction mixture was heated at reflux for 16 h, crushed ice was added, and the precipitate was collected via filtration and crystallized using ethanol. Yield: 78%, mp: 97–99 °C. ^1^H-NMR (DMSO-d_6_) δ 2.50 (s, 4H, 2 × CH_2_), 3.11 (s, 2H, CH_2_), 3.63 (s, 4H, 2 × CH_2_), 4.23 (s, 2H, CH_2_), 7.52 (d, 2H, J = 8.69, ArH), 7.58 (d, 2H, J = 8.85, ArH), 9.71 (s, 2H, NH, NH).

#### 4.1.5. General Procedure for Triazole Thiol (**7a**–**f**) Preparation

A solution of previously prepared indolyl-3-carbonyl-*N*-substituted phenyl thiosemicarbazides (**6a**–**f**, 10 mmol) in 2N NaOH (4 mL) was refluxed for 3 h [49]. Water was added, and the solution was carefully neutralized with diluted HCl. The corresponding precipitate was filtered, dried, and recrystallized from ethanol.

##### 4-(4-Fluorophenyl)-5-(1H-indol-3-yl)-4H-1,2,4-triazole-3-thiol (**7a**)

Yield: 62%, mp: 223–225 °C. ^1^H-NMR (DMSO-d_6_) δ 6.46 (d, 1H, J = 2.53, ArH), 7.18 (m, 2H, ArH), 7.42–7.48 (m, 3H, ArH), 7.51–7.55 (m, 2H, ArH), 8.06 (d, 1H, J = 7.50, ArH), 11.45 (s, 1H, NH), 13.94 (s, 1H, SH).

##### 4-(Methoxyphenyl)-5-(1H-indol-3-yl)-4H-1,2,4-triazole-3-thiol (**7b**)

Yield 56%, mp: 210–212 °C. ^1^H-NMR (DMSO-d_6_) δ 3.86 (s, 3H, OCH_3_), 6.40 (d, 1H, J = 2.85, ArH), 7.12–7.22 (m, 4H, ArH), 7.35 (d, 2H, J = 8.91, ArH), 7.43 (d, 1H, J = 7.84, ArH), 8.09 (d, 1H, J = 7.84, ArH), 11.52 (s, 1H, NH), 13.80 (s, 1H, SH).

##### 5-(1H-Indol-3-yl)-4-methylphenyl-4H-1,2,4-triazole-3-thiol (**7c**)

Yield 60%, mp: 208–210 °C. ^1^H-NMR (DMSO-d_6_) δ 2.45 (s, 3H, CH_3_), 6.38 (d, 1H, J = 2.84, ArH), 7.14–7.23 (m, 2H, ArH), 7.32 (d, 2H, J = 8.18, ArH), 7.43 (d, 3H, J = 8.18, ArH), 8.08 (d, 1H, J = 8.18, ArH), 11.40 (s, 1H, NH), 13.93 (s, 1H, SH).

##### 4-(3-Chlorophenyl)-5-(1H-indol-3-yl)-4H-1,2,4-triazole-3-thiol (**7d**)

Yield: 71%, mp: 170–172 °C. ^1^H-NMR (DMSO-d_6_) δ 6.49 (d, 1H, J = 2.49, ArH), 7.19 (m, 2H, ArH), 7.45 (t, J = 7.50 2H, ArH), 7.64 (t, J = 6.50, 1H, ArH), 7.68 (d, 1H, J = 1.25, ArH), 7.7 (d, 1H, J = 1.25, ArH), 8.0 (d, 1H, J = 7.97, ArH), 11.49 (s, 1H, NH), 13.9 (s, 1H, SH).

##### 4-(4-Chlorophenyl)-5-(1H-indol-3-yl)-4H-1,2,4-triazole-3-thiol (**7e**)

Yield: 67%, mp: 165–167 °C. ^1^H-NMR (DMSO-d_6_) δ 6.51 (d, 1H, J = 2.48, ArH), 7.19 (m, 2H, ArH), 7.45 (d, 1H, J = 8.14, ArH), 7.51 (d, 2H, J = 8.50, ArH) 7.68 (d, 2H, J = 8.14, ArH), 8.04 (d, 1H, J = 7.79, ArH), 11.40 (s, 1H, NH), 13.96 (s, 1H, SH).

##### 4-(3,4-Dichlorophenyl)-5-(1H-indol-3-yl)-4H-1,2,4-triazole-3-thiol (**7f**)

Yield: 63%, mp: 198–200 °C. ^1^H-NMR (DMSO-d_6_) δ 6.62 (d, 1H, J = 2.85, ArH), 7.17 (m, 2H, ArH), 7.46 (d, 1H, J = 8.05, ArH), 7.5 (d, 1H, J = 8.56, ArH), 7.87 (d, 1H, J = 8.67, ArH), 7.92 (s, 1H, ArH), 8.0 (d, 1H, J = 7.97, ArH), 11.63 (s, 1H, NH), 14.0 (s, 1H, SH).

#### 4.1.6. General Procedure of S Alkylation of Triazole Thiol (**U1**–**6**)

*N*-[4-(2-chloroacetylamino)phenyl]-2-morpholin-4-yl-acetamide (**5**, 10 mmol) was added to a mixture of 4-(substituted-phenyl)-5-(1H-indol-3-yl)-4H-1,2,4-triazole-3-thiol (**7a**–**f**, 10 mmol) in ethanol and KOH (10 mmol) and stirred for 16 h at room temperature. The produced precipitate was then filtered, dried, and recrystallized from ethanol to obtain the corresponding *S*-alkyl triazole thiol **U1**–**6**.

##### 2-(4-(4-Fluorophenyl)-5-(1H-indol-3-yl)-4H-1,2,4-triazol-3-ylthio)-*N*-(4-(2 morpholinoacetamido)phenyl)acetamide (**U1**)

Yield: 65%, mp: 208–210 °C. ^1^H-NMR (DMSO-d_6_) δ 2.14 (s, 4H, 2 × CH_2_), 3.14 (s, 4H, 2 × CH_2_), 3.63 (s, 2H, CH_2_), 4.15 (s, 2H, CH_2_), 6.60 (d, 1H, J = 2.12, ArH), 7.19 (m, 2H, ArH), 7.45 (d, 1H, J = 6.37, ArH), 7.50–7.56 (m, 4H, ArH), 7.60–7.66 (m, 4H, ArH), 8.2 (d, 1H, J = 7.79, ArH), 9.73 (s, 1H, NH), 10.32 (s, 1H, NH), 11.35 (s, 1H, NH). MS analysis for C_30_H_28_FN_7_O_3_S: Calcd mass: 585.65, found (*m*/*z*, M+): 586.16.

##### 2-(5-(1H-Indol-3-yl)-4-(4-methoxyphenyl)-4H-1,2,4-triazol-3-ylthio)-*N*-(4-(2-morpholinoacetamido)phenyl)acetamide (**U2**)

Yield: 65%, mp: 173–175 °C. ^1^H-NMR (DMSO-d_6_) δ 2.12 (s, 4H, 2 × CH_2_), 3.11 (s, 4H, 2 × CH_2_), 3.60 (s, 2H, CH_2_), 3.86 (s, 3H, OCH_3_), 4.11 (s, 2H, CH_2_), 6.54 (d, 1H, J = 2.21, ArH), 7.15 (m, 4H, ArH), 7.41–7.51 (d, 4H, J = 8.50, ArH), 7.60 (d, 3H, J = 8.50, ArH), 8.20 (d, 1H, J = 5.66, ArH), 9.68 (s, 1H, NH), 10.31 (s, 1H, NH), 11.30 (s, 1H, NH). ^13^C-NMR (DMSO-d_6_): δ 34.60, 53.08, 53.38, 59.80, 66.65, 106.58, 112.50, 116.01, 119.50, 120.06, 120.29, 120.56, 121.59, 122.74, 127.29, 127.31, 127.66, 131.94, 134.52, 134.61, 136.60, 148.85, 159.61, 159.87, 166.61, 169.58. MS analysis for C_31_H_31_FN_7_O_4_S: Calcd mass 597.69, found (*m*/*z*, M+): 598.22.

##### 2-(5-(1H-Indol-3-yl)-4-methylphenyl-4H-1,2,4-triazol-3-ylthio)-*N*-(4-(2-morpholinoacetamido)phenyl)acetamide (**U3**)

Yield: 67%, mp: 250–252 °C. ^1^H-NMR (DMSO-d_6_) δ 2.10 (s, 4H, 2 × CH_2_), 2.41 (s, 3H, CH_3_), 3.09 (s, 4H, 2 × CH_2_), 3.57 (s, 2H, CH_2_), 4.09 (s, 2H, CH_2_), 6.50 (d, 1H, J = 2.12, ArH), 7.19 (m, 2H, ArH), 7.45 (dd, 5H, J = 6.37, 7.15 ArH), 7.52 (d, 2H, J = 6.90, ArH), 7.64 (d, 2H, J = 6.90, ArH), 8.2 (d, 1H, J = 7.79, ArH), 9.73 (s, 1H, NH), 10.39 (s, 1H, NH), 11.43 (s, 1H, NH). ^13^C-NMR (DMSO-d_6_): δ 21.07, 34.60, 53.08, 59.80, 66.65, 106.60, 112.50, 120.06, 120.29, 120.56, 121.59, 122.74, 125.82, 127.29, 127.31, 130.56, 134.52, 134.61, 134.79, 136.60, 137.50, 148.90, 159.65, 166.61, 169.58. MS analysis for C_31_H_31_FN_7_O_3_S: Calcd mass 581.69, found (*m*/*z*, M+): 582.18.

##### 2-(4-(3-Chlorophenyl)-5-(1H-indol-3-yl)-4H-1,2,4-triazol-3-ylthio)-*N*-(4-(2-morpholinoacetamido)phenyl) acetamide (**U4**)

Yield: 45%, mp: 220–222 °C. ^1^H-NMR (DMSO-d_6_) δ 2.16 (s, 4H, 2 × CH_2_), 3.17 (s, 4H, 2 × CH_2_), 3.68 (s, 2H, CH_2_), 4.18 (s, 2H, CH_2_), 6.50 (d, 1H, J = 2.30, ArH), 7.15 (m, 2H, ArH), 7.45 (d, 1H, J = 6.37, ArH), 7.54–7.57 (m, 4H, ArH), 7.62–7.66 (m, 3H, ArH), 7.74 (t, J = 6.75, 1H, ArH), 8.2 (d, 1H, J = 7.79, ArH), 9.63 (s, 1H, NH), 10.52 (s, 1H, NH), 11.34 (s, 1H, NH). MS analysis for C_30_H_28_ClN_7_O_3_S: Calcd mass: 601.17, found (*m*/*z*, M+): 602.20.

##### 2-(4-(4-Chlorophenyl)-5-(1H-indol-3-yl)-4H-1,2,4-triazol-3-ylthio)-*N*-(4-(2-morpholinoacetamido)phenyl)acetamide (**U5**)

Yield 65%, mp: 208–210 °C. ^1^H-NMR (DMSO-d_6_) δ 2.15 (s, 4H, 2 × CH_2_), 3.16 (s, 4H, 2 × CH_2_), 3.64 (s, 2H, CH_2_), 4.18 (s, 2H, CH_2_), 6.58 (d, 1H, J = 2.42, ArH), 7.14 (m, 2H, ArH), 7.40 (d, 1H, J = 8.37, ArH), 7.52–7.54 (m, 4H, ArH), 7.68 (d, 2H, J = 7.50, ArH), 7.62–7.64 (m, 2H, ArH), 8.2 (d, 1H, J = 7.79, ArH), 9.73 (s, 1H, NH), 10.30 (s, 1H, NH), 11.20 (s, 1H, NH). MS analysis for C_30_H_28_ClN_7_O_3_S: Calcd mass: 601.17, found (*m*/*z*, M+): 601.98.

##### 2-[4-(3,4-Dichlorophenyl)-5-(1H-indol-3-yl)-4H-1,2,4-triazol-3-ylthio-*N*-[4-(2-morpholinoacetamido)phenyl acetamide (**U6**)

Yield: 57%, mp: 218–220 °C. ^1^H-NMR (DMSO-d_6_) δ 2.10 (s, 4H, 2 × CH_2_), 3.04 (s, 4H, 2 × CH_2_), 3.12 (s, 2H, CH_2_), 4.07 (s, 2H, CH_2_), 6.58 (d, 1H, J = 2.12, ArH), 7.18–7.21 (m, 2H, ArH), 7.46 (d, 1H, J = 8.05, ArH), 7.64 (t, J = 7.1, 1H, ArH), 7.70–7.74 (m, 3H, ArH), 7.82–7.84 (m, 2H, ArH), 7.92 (s, 1H, ArH), 8.0 (d, 1H, J = 7.97, ArH), 9.73 (s, 1H, NH), 10.34 (s, 1H, NH), 11.63 (s, 1H, NH). MS analysis for C_30_H_27_Cl_2_N_7_O_3_S: Calcd mass: 635.13, found (*m*/*z*, M+): 636.78.

### 4.2. Biology

#### 4.2.1. Cell Culture and Maintenance

MDA-231 triple-negative breast cancer cells, MCF-7 breast cancer cells, and A549 adenocarcinomic human alveolar basal epithelial cells were maintained in Roswell Park Memorial Institute media (RPMI 1640, Sigma-Aldrich, St. Louis, MO, USA) supplemented with 10% foetal bovine serum and 1% penicillin/streptomycin. HDF human dermal normal fibroblast cells were cultured in Dulbecco’s Modified Eagle Medium (DMEM, Sigma-Aldrich, Eppelheim, Germany). MCF-7 breast cancer cell line was obtained from Cell Lines Service (CLS; Eppelheim, Germany). MDA-MB-231 breast cancer cell line and A549 cancer cells were obtained from the European Collection of Authenticated Cell Cultures (ECACC; Gillingham UK). Cell lines were maintained at 37 °C in a humidified incubator with 5% CO_2_.

#### 4.2.2. Cytotoxicity Assay

The antiproliferative activity of compounds **U1**–**6** was assessed using the 3-[4,5-dimethylthiazol-2-yl]-2,5-diphenyl tetrazolium bromide (MTT) assay, as previously described [40,64]. Cancer cell lines were seeded in 96-well flatbottom plates at a density of 10^4^ cells/well for 24 h. The cells were subsequently treated with **U1**–**6** compounds at a screening concentration of 50 µM and incubated at 37 °C for 48 h in a humidified incubator containing 5% CO_2_. Serial dilution concentrations ranged from 1, 10, 25, 50, to 100 µM and were used for IC_50_ calculations of each compound for the three cancer cell lines. DMSO was used as a negative vehicle control (1% concentration). The culture media were then removed and incubated for 2 h with fresh 200 µL culture media containing 5 mg/mL MTT. After removing the culture media and dissolving the produced formazan crystals in 100 µL DMSO, they were incubated for another 30 min at 37 °C. The produced colour was measured at 570 nm using a microplate reader (Thermo-Scientific, Vantaa, Finland). The GraphPad Prism version 9.1 software was used to generate plots of absorbance versus the compound’s concentration. Three independent repeat experiments were performed for each concentration to ensure reproducibility. IC_50_ values were obtained from nonlinear regression plots of absorbance versus the log concentration of the tested compounds using the GraphPad Prism software version 5.0 (San Diego, CA, USA).

#### 4.2.3. Cell Cycle Assay

Most active compound **U2** was applied to MCF7 cells for 24 h at its IC_50_ concentration. DMSO vehicle was used as a negative control (1% concentration). The cells were collected, centrifuged, and fixed in 70% ethanol on ice for 20 min. The fixed cells were incubated for 1 h at room temperature with staining solution (50 mg/mL propidium iodide PI, 0.05% Triton X-100, 0.1 mg/mL RNaseA). A Gallios flow cytometer (Beckman Coulter, Brea, CA, USA) was used to measure the cell cycle proportion [70].

#### 4.2.4. Apoptosis Assay

The Annexin-V-FITC apoptosis detection kit (Catalog # k101-25, Biovision, Mountain View, USA) was used for the apoptosis experiment according to previously published data. MCF-7 cells were treated with **U2** compound for 24 h. The negative control was 0.1% DMSO. The cells were collected via centrifugation and resuspended in 500 µL of 1X Binding Buffer. Treated and control cells were stained with 5 µL Annexin V-FITC and 5 µL propidium iodide (PI 50 mg/mL) and then incubated at room temperature for 5 min in the dark. Fluorescence-activated cell sorting (FACS) was quantified via flow cytometry. Flow cytometry (excision at 488 nm; emission at 530 nm) was used to examine Annexin V-FITC binding and PI staining using the FITC signal detector and a phycoerythrin emission signal detector [71].

#### 4.2.5. Bim Enzyme-Linked Immunosorbent Assay (ELISA)

A Bim-binding assay (Thermo-Scientific, Catalogue No # BMS244-3, Vienna, Austria) was performed according to previously published data [48,49]. PBS solution containing 0.05% Tween-20 was used to wash a streptavidin-coated 96-well plate. For immobilisation, biotinylated Bim peptide (residues 81–106) was diluted in SuperBlock blocking solution and applied to each well. After incubation, the plate was washed with 0.5% BSA in PBS with Tween-20 solution. In PBS, test compounds were treated with His-tagged Bcl-2 protein and incubated for 1 h and then transferred to the wells containing the immobilised Bim peptide. After another wash, each well received anti-His antibody with horseradish peroxidase enzyme. Following incubation and washing, *O*-phenylenediamine and hydrogen peroxide solution were added for colour production. A plate reader (Thermo-Scientific, Vantaa, Finland) was used to measure the optical density at 450 nm. The reduction in Bim affinity was measured using a nonlinear regression curve (GraphPad Prism. Version 5.0), which was also utilized to generate the IC_50_ value of Bcl-2 inhibition.

### 4.3. Computational Modelling 

All computational work was carried out using Schrödinger suite 12.7, available at www.schrödinger.com, accessed on 5 April 2023, and using the Maestro graphical user interface software.

#### 4.3.1. Protein and Ligand Preparation

The protein data bank (https://www.rcsb.org, accessed on 1 April 2023), was used to obtain the 3D crystal structure of human Bcl-2 (PDB ID: 4AQ3). The protein was prepared and refined using the Protein Preparation Wizard Maestro. Water molecules that were crystallographically larger than 5 A° were eliminated. At pH 7.3, all the missing hydrogen atoms were added to the protein for correct ionization using the EPIK module [72,73]. The tautomerization state and bond order of amino acid residues were assigned. Water molecules with three hydrogen bonds to non-waters were eliminated. Finally, to alleviate steric conflicts, energy was minimized using OPLS4 [74,75]. Ligand compounds were built and energy minimized using the LigPrep tool [76,77], and the ligand 2D structure was transformed into a 3D structure [78].

#### 4.3.2. Grid Generation and Molecular Docking

In grid generation, a ligand with the crystal structure of human Bcl-2 (PDB ID: 4AQ3) was utilized. For docking investigations, a grid box was constructed at the active site’s centroid, and the active site was defined around the native ligand of the Bcl-2 (PDB ID: 4AQ3) crystal structure. The prepared ligands were docked within the grid-generated Bcl-2 (PDB: 4AQ3) binding site using the standard precision (SP) mode of Glide without any limitations [79,80]. The visual inspection of the interaction indicated the affinity of the docked ligands to the binding site.

#### 4.3.3. Pharmacokinetics Predication 

Using the Swiss-ADME website (http://www.swissadme.ch/), the bioavailability radar of the most active substance, **U2**, was projected versus Bcl-2 small-molecule inhibitor ABT-263. Structure sketches were converted into the SMILES format [81,82]. The bioavailability radar indicated desired features such as size, solubility, saturation, polarity, lipophilicity, and flexibility [83].

### 4.4. Statistical Analysis

GraphPad Prism (version 9.1.0) was used for data analysis using one-way analysis of variance (ANOVA) and the multiple comparison test. The data were represented by the mean SEM of three independent replicates. *p <* 0.05 was used as the statistical significance level: * represents a *p*-value of <0.05, ** represents a *p*-value of <0.01, *** represents a *p*-value of <0.001, **** represents a *p*-value of <0.0001.

## 5. Conclusions

This study investigated the effects of designed indole-based derivatives **U1**–**6** on Bcl2-expressing cancer cell growth, apoptosis, and cell cycle progression. Out of the designed compounds, compound U2, with a 4-methoxy substitution, demonstrated potent inhibitory effects against cell growth through the induction of apoptosis. In comparison with untreated control cells, the **U2** compound exhibited a significant 43-fold increase in early apoptosis and a remarkable 111-fold increase in late apoptosis. Moreover, the compound induced cell cycle arrest at the G1/S phase. These findings highlight the promising anticancer activity of U2. Further research and development of **U2** may lead to novel strategies for combating cell-proliferation-related diseases.

## Data Availability

No data available.

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
