# Peer review of "Design, Synthesis, and Potent Anticancer Activity of Novel Indole-Based Bcl-2 Inhibitors"

_ijms, 2023, doi:10.3390/ijms241914656_

Round 1
Reviewer 1 Report
The paper entitled “Design, synthesis, and potent anticancer activity of novel indole-based Bcl-2 inhibitors” describes the synthesis of novel indole-based compounds and their biological evaluation as anti-cancer agents. Further experiments, such as ELISA and molecular docking, identified BCL-2 as target. The work could be depicted as routine medicinal chemistry, without any significant novelty in the chemistry part although the biological section shows interesting results, with some of the compounds demonstrating powerful inhibitory activity comparable to the reference. In particular, compound U2 showed to be the most interesting hit for further development. Biological part furnished compelling outcomes, while I have many concerns about publication of the work in IJMS, especially for chemistry section, which needs to be improved consistently. I consider major revision overall and I will support publication only if the following key-points will be clarified.
- The introduction contains a description of the target BCL-2 but no mention of the importance of the indole core in medicinal chemistry and drug discovery has been made, since “indole-based” has been included in the title of the work. I think that would be worth to mention some biologically relevant indole scaffold as reference, I can suggest some recent papers: antiproliferative activity (Bioorganic Chemistry, 2022, 129: 106213, ChemistrySelect, 2022, 7.34: e202202361), antiviral activity (Organic & Biomolecular Chemistry, 2023, 21 (18), 3811-3824, Cell Death Discovery, 2022, 8.1: 491, Journal of Molecular Structure, 2022, 1261: 132808) and general biologically active indoles (Pharmacological Reports, 2022, 74.4: 570-582).
- Experimental Chemistry: “the identity and purity of synthesised compounds were monitored on thin-layer chromatography (TLC)” should be removed, TLC is not in any cases a sufficient analytical method to identify and characterize a chemical compound. Also purity cannot be assured by TLC. Purity of the compounds should be guaranteed by HPLC analysis or at least elemental analysis. Moreover, all the new compounds need to be fully characterized by HNMR, CNMR and MS as stated by guidelines of IJMS. In my opinion, it is not possible to publish a paper with newly synthesized compounds without any of these structural data. Please insert 13CNMR data for all the reported new compounds.
- Synthesis of the title compounds: I do not think that mixing DCM with potassium carbonate represents a “solution”, at least a "suspension". “R” substituent needs to be depicted in the Scheme.
- Discussion section does not fit with the “style” of the journal and represents a trivial repetition. Results section could furnish all the relevant information of the paper.
- A copy of 1HNMR and 13CNMR spectra needs to be included in the Supplementary files, to prove identity and purity of the target compounds.
Author Response
Comment 1# The introduction contains a description of the target BCL-2 but no mention of the importance of the indole core in medicinal chemistry and drug discovery has been made, since “indole-based” has been included in the title of the work. I think that would be worth to mention some biologically relevant indole scaffold as reference, I can suggest some recent papers: antiproliferative activity (Bioorganic Chemistry, 2022, 129: 106213, ChemistrySelect, 2022, 7.34: e202202361), antiviral activity (Organic & Biomolecular Chemistry, 2023, 21 (18), 3811-3824, Cell Death Discovery, 2022, 8.1: 491, Journal of Molecular Structure, 2022, 1261: 132808) and general biologically active indoles (Pharmacological Reports, 2022, 74.4: 570-582).
Response 1# The introduction was modified and updated the biological activity of indole scaffold in the manuscript as below:
Indole scaffolds play a crucial part in cancer cell targeting particularly because the nitrogen atom in indole can form hydrogen bonds with the biological targets, enhancing their potential activity [32-34]. Several research groups demonstrated the incorporation of the indole scaffold and structural optimization to target cancer cells [34-37]. Obatoclax is indole based Bcl-2 inhibitor [38] that inspired the design of several indole-based compounds to bind directly to the BCL-2 protein and suppress its anti-apoptotic action [39, 40]. It also influences mitochondrial apoptotic pathway by changing the permeability of the mitochondrial membrane, releasing pro-apoptotic proteins, and inducing apoptosis [41, 42]. Other indole-based compounds have caused the production of reactive oxygen species [43] and caused cancer cells to enter cell cycle arrest [44], halting uncontrolled proliferation and triggering apoptosis [45].
Comment 2#: Experimental Chemistry: “the identity and purity of synthesised compounds were monitored on thin-layer chromatography (TLC)” should be removed, TLC is not in any cases a sufficient analytical method to identify and characterize a chemical compound. Also purity cannot be assured by TLC. Purity of the compounds should be guaranteed by HPLC analysis or at least elemental analysis. Moreover, all the new compounds need to be fully characterized by HNMR, CNMR and MS as stated by guidelines of IJMS. In my opinion, it is not possible to publish a paper with newly synthesized compounds without any of these structural data. Please insert 13CNMR data for all the reported new compounds.
Response 2#: The sentence concerning purity monitoring by TLC was removed from the manuscript and the 13C NMR of the most active compounds U2 and U3 were included as the most relevant final compounds for further development.
Comment 3#: Synthesis of the title compounds: I do not think that mixing DCM with potassium carbonate represents a “solution”, at least a "suspension". “R” substituent needs to be depicted in the Scheme.
Response 3#: The sentence was corrected in the manuscript. The R substituent was added to the Scheme and updated in the manuscript.
Comment 4#: Discussion section does not fit with the “style” of the journal and represents a trivial repetition. The Results section could furnish all the relevant information of the paper.
Response 4#: The discussion section was updated by commentary on future studies in the manuscript. The results section was updated to cover missing information in the docking section.
Comment 5#: A copy of 1H NMR and 13C NMR spectra needs to be included in the Supplementary files, to prove identity and purity of the target compounds.
Response 5#: The 1H NMR and 13C NMR spectra of compounds U2 and U3 as the most active compounds were added as supplementary files to further demonstrate their purity.

Reviewer 2 Report
This manuscript by Almehdi et al. described the design, synthesis, and anticancer activity of a series of indole-based Bcl-2 inhibitors. Bcl-2 is an important protein target for cancers since it’s a key regulator of apoptosis. Reported small molecule inhibitors of Bcl-2 include Obatoclax and Venetoclax. Venetoclax is an FDA-approved selective Bcl-2 inhibitor. This manuscript reported a series of indole-based Bcl-2 inhibitors with modest affinity against Bcl-2. ELISA was used to confirm the binding affinity. Docking study was followed to validate the binding mode of the inhibitors against Bcl-2. In addition, cytotoxicity assay, cell cycle assay, and apoptosis assay were utilized to validate the anticancer activity. Following points need to be addressed:
1. Line 49, reference 9: There are more Bcl-2 inhibitors in clinical trials after 2013. Update the reference.
2. Figure 1: The bond angles in the ester of Venetoclax and CF3 group of Navitoclax were wrong.
3. Experimental section: Experimental details missed, such as % of DMSO in 2.2.2 and 2.2.3.
4. 2.3.1. Protein and ligand prep: EPIK module instead of EPIC. Cite the original reference to support your claim. For example: energy was minimized using OPLS4, cite “OPLS3: A force field providing broad coverage of drug-like small molecules and protein”. There are a few other references needed to be updated, such as reference 41 and reference 45.
5. Table 1 and Figure 4: Did you use SD or SEM? It’s inconsistent in the table and figure captions.
6. Have you ever tested other potent Bcl-2 inhibitors using ELISA? The curves didn’t reach the bottom plateau due to the low affinity of the inhibitors. Have you ever tried to increase the top concentration a little bit? In the discussion section, the authors mentioned that Venetoclax resistance limited the use of Venetoclax. Dual Bcl-2 and MCL-1 inhibitors could be a potential strategy. Have you tested the leading molecule against other anti-apoptotic members (BCL-XL and MCL-1)?
7. Figure 7: No error bars on A and D. Have you ever used other Bcl-2 inhibitors as positive controls? There are a few reports talking about Bcl-2 inhibitors inducing G0/G1 arrest. In reference 68, it also mentioned “24 h of exposure to OBAT at a concentration of 500 nM resulted in a marked accumulation of cells in the G0/G1 phase accompanied with a significant reduction of cells in the G2/M phase of cell cycle”. It will be easier to interpret the data with a positive control in the cell cycle and apoptosis assays.
8. Figure 8: Show the distances of the potential hydrogen bonding interactions and pi-pi stacking interactions. Hide the side chains of other residues to simplify the figures.
9. Table 3: It will be easier to show the potential interactions with 2D diagrams.
10. Line 457: Ref 56 focused dual Bcl-2/Bcl-XL inhibitors, selective Bcl-2 inhibitors, and selective MCL-1 inhibitors. Need a different reference to support your claim.
Minor points:
1. Inconsistent significant figures in 2.1. Chemistry.
2. Line 115: remove the extra period.
3. Line 380: missing period.
4. Line 394: remove the extra period.
5. Line 422: stacking instead of staking.
6. Be consistent in citation format.
Author Response
Comment 1#: Line 49, reference 9: There are more Bcl-2 inhibitors in clinical trials after 2013. Update the reference.
Response 1#: The reference already updated in the manuscript with recent reference 2023 (reference [10] in the revised manuscript).
Comment 2#: Figure 1: The bond angles in the ester of Venetoclax and CF3 group of Navitoclax were wrong.
Response 2#: The figure was modified and updated in the manuscript.
Comment 3#: Experimental section: Experimental details missed, such as % of DMSO in 2.2.2 and 2.2.3.
Response 3#: The percentage of the DMSO was updated now in the manuscript.
Comment 4#: 2.3.1. Protein and ligand prep: EPIK module instead of EPIC. Cite the original reference to support your claim. For example: energy was minimized using OPLS4, cite “OPLS3: A force field providing broad coverage of drug-like small molecules and protein”. There are a few other references needed to be updated, such as reference 41 and reference 45.
Response 4#: The references were updated in the manuscript and EPIC was replaced by EPIK module.
Comment 5#: Table 1 and Figure 4: Did you use SD or SEM? It’s inconsistent in the table and figure captions.
Response 5#: Thanks, it was corrected now as SEM in both Table 1 and Figure 4.
Comment 6#: Have you ever tested other potent Bcl-2 inhibitors using ELISA? The curves didn’t reach the bottom plateau due to the low affinity of the inhibitors. Have you ever tried to increase the top concentration a little bit? In the discussion section, the authors mentioned that Venetoclax resistance limited the use of Venetoclax. Dual Bcl-2 and MCL-1 inhibitors could be a potential strategy. Have you tested the leading molecule against other anti-apoptotic members (BCL-XL and MCL-1)?
Response 6#: Thanks for the useful suggestion that will we incorporate into our future research. Considering Reviewer #2 comments, the following text has been added to end of the Discussion section: “A future priority will be to test the activity on anti-apoptotic members as (BCL-XL and MCL-1)”.
Comment 7#: Figure 7: No error bars on A and D. Have you ever used other Bcl-2 inhibitors as positive controls? There are a few reports talking about Bcl-2 inhibitors inducing G0/G1 arrest. In reference 68, it also mentioned “24 h of exposure to OBAT at a concentration of 500 nM resulted in a marked accumulation of cells in the G0/G1 phase accompanied with a significant reduction of cells in the G2/M phase of cell cycle”. It will be easier to interpret the data with a positive control in the cell cycle and apoptosis assays.
Response 7#: The error bar now added to the figure A and D. We only examine the activity of U2 on the cell cycle and apoptosis assays without interpreting the result with a positive control, which we will consider in a future study. In the light of this context, the following text has been added to the manuscript “A future study will be carried to compare the activity of our designed compound compared to further positive control compounds in the cell cycle and apoptosis assays.”
Comment 8#: Figure 8: Show the distances of the potential hydrogen bonding interactions and pi-pi stacking interactions. Hide the side chains of other residues to simplify the figures.
Response 8#: The figure was simplified as requested the distance was measured and updated in the manuscript.
Comment 9#: Table 3: It will be easier to show the potential interactions with 2D diagrams.
Response 9#: The 2D diagram was added to the manuscript and the table showed the Aromatic H-bond and hydrophobic interaction that were not shown in the figure.
Comment 10#: Line 457: Ref 56 focused dual Bcl-2/Bcl-XL inhibitors, selective Bcl-2 inhibitors, and selective MCL-1 inhibitors. Need a different reference to support your claim.
Response 10#: New references already updated in the manuscript.
Minor points:
Comment 1#: Inconsistent significant figures in 2.1. Chemistry.
Response 1#: It has been modified as requested and updated in the manuscript.
Comment 2#: Line 115: remove the extra period.
Response 2#: It has been modified as requested and updated in the manuscript.
Comment 3#: Line 380: missing period.
Response 3#: It has been modified as requested and updated in the manuscript.
Comment 4#: Line 394: remove the extra period.
Response 4#: It has been modified as requested and updated in the manuscript.
Comment 5#: Line 422: stacking instead of staking.
Response 5#: It has been modified as requested and updated in the manuscript.
Comment 6#: Be consistent in citation format.
Response 6#: The format updated to IJMS format.

Round 2
Reviewer 1 Report
Dear authors,
thank you for considering my comments. Good luck for your publication.
Reviewer 2 Report
Thanks for addressing all the comments. I still think it will be better to include the positive controls for some of the cell based experiments. Look forward to the followup studies.